

# Evaluation of the effects of whitening toothpaste containing nanohydroxyapatite on surface roughness and color change in restorative materials

Burak Dayı and Fikri Öcal

Department of Restorative Dentistry, Faculty of Dentistry, Inonu University, Malatya, Turkey

## ABSTRACT

**Background:** The effects of whitening toothpastes containing nanohydroxyapatite on the surfaces of restorative materials are not well known. This study evaluated the changes in surface roughness and color of coffee-stained restorative materials after brushing with nanohydroxyapatite and other whitening toothpastes.

**Methods:** Disc-shaped specimens were formed using microhybrid, nanohybrid, and supra-nano-filled composite ($n = 30$) and stained with a coffee solution. A brushing simulation was applied with toothpastes containing nanohydroxyapatite (Dentiste Plus White), perlite (Signal White System) and hydrogen peroxide (Colgate Optic White) for 7, 15, and 30 days. Color and surface roughness measurements were taken before and after brushing. Color change ($\Delta E_{00}$) was calculated using the CIEDE2000 system.

**Results:** Supra-nano-filled composite-Dentiste Plus White, supra-nano-filled composite-Colgate Optic White, and nanohybrid composite-Signal White System were the groups with the greatest color change observed on the 7th, 15th, and 30th days, respectively. The smallest color change was seen in the microhybrid composite-Signal White System, microhybrid composite-Dentiste Plus White, and nanohybrid composite-Dentiste Plus White groups on the 7th, 15th, and 30th days, respectively. No significant surface roughness changes were observed in the nanohybrid composite-Dentiste Plus White, supra-nano-filled composite-Colgate Optic White, supra-nano-filled composite-Dentiste Plus White, or supra-nano-filled composite-Signal White System groups.

**Conclusions:** Whitening toothpaste containing nanohydroxyapatite provided a high degree of color change in the short term and did not create significant surface roughness in nanohybrid or supra-nano-filled composites.

Corresponding author
Burak Dayı, bdayi70@hotmail.com

# INTRODUCTION

Today, patient demand for esthetic restorations has led to an increase in the use of resin composites and studies on these materials. To provide an aesthetic appearance in restorative dentistry, restorations resembling natural teeth should be applied, and this similarity should be maintained for a long time. In this context, the color stability of the

composite restoration and the detection of discoloration are important criteria in evaluating material success (*Roselino et al., 2013*). One of the important factors required for composite restoration to preserve its color for a long time is the maintenance of surface smoothness (*Kocaagaoglu et al., 2017*). Tooth brushing time, certain toothpaste types, and applied force can negatively affect the long-term durability of the surface shine of composite restorations (*de Moraes Rego Roselino et al., 2019*).

Since resin composites were first developed, efforts have been made to enhance their clinical efficacy. Whereas resin composite research is mostly focused on the creation of novel monomers, investigations on filler content concentrate on particle size and silanization (*Atai et al., 2004*; *Lu et al., 2005*). Nanotechnology examines materials in the range of about 0.1–100 nm and is known as the manufacture and manipulation of these structures using different physical or chemical processes (*Beun et al., 2007*). Whereas the particle size in hybrid composites is 8–30 μm and around 0.7–3.6 μm in microhybrid composites, restorative materials with particle sizes ranging from 5–100 nm have been recently developed (*Sivakumar et al., 2016*; *Moszner & Klapdohr, 2004*). Composites containing supra-nano particles have been created using the sol-gel process, which adjusts the diameter of the fillers while also changing their refractive index (*Perez et al., 2016*). The manufacturer states that the particles in the filler content are spherical, allowing it to maintain its brightness and exhibit a realistic opalescence (*Tokuyama Dental, 2011*).

Toothpastes with whitening properties used to clean the teeth and make them look whiter may contain abrasives such as hydrated silica, calcium carbonate, calcium pyro-phosphate, perlite, and sodium bicarbonate (*Epple, Meyer & Enax, 2019*). In addition to abrasives, toothpastes containing chemicals (*e.g.*, phosphates) and dyes (*e.g.*, blue covariate) are also available (*Sarembe et al., 2020*). Research has focused on toothpastes that whiten while causing minimal abrasion to the surface of teeth and restorative materials (*Vaz et al., 2019*; *Schlafer et al., 2021*). Whitening toothpastes containing nano-hydroxyapatite (n-HAP), a calcium phosphate derivative, is among these.

Hydroxyapatite (HAP) is morphologically similar to the apatite crystal found in human tooth enamel. In addition to the biocompatible and biomimetic properties of this molecule, its whitening properties have attracted the attention of dentists (*Shang, Kaisarly & Kunzelmann, 2022*). Many researchers have applied hydroxyapatite to teeth in various forms and stated that it has a whitening effect (*Onuma, Yamagishi & Oyane, 2005*; *Kim et al., 2006*). *Jin et al. (2013)* stated that HAP whitens not only because it is white, but also because a freshly formed thin coating of HAP is thought to help diffuse reflections on the tooth surface, making the teeth look brighter.

Knowing the discoloration and abrasive effects of whitening toothpastes with different active ingredients, which are frequently preferred by patients for teeth whitening, on teeth and composite restorations can help physicians to recommend a suitable whitening toothpaste for the appropriate patient. In this way, patients may minimize the roughness that may occur on the restoration surfaces and the negative effects of this situation, while using an effective whitening toothpaste for removing and preventing discoloration in composite restorations. There are many studies in the literature on the effects of whitening toothpastes on teeth and composite restorations (*Roselino et al., 2013*; *de Moraes Rego*

**Table 1 Restorative materials used in the study.**

| Composite | Type | Compound | Manufacturer |
|---|---|---|---|
| Filtek Z250 | Microhybrid | Matrix: Bis GMA, Bis EMA, UDMA, TEGDMA<br>Filler: Zirconia/Silica<br>Filler size: 0.01–3.5 μm | 3M ESPE, St. Paul, MN, USA |
| Polofil NHT | Nanohybrid | Matrix: Bis GMA, UDMA, TEGDMA<br>Filler: Glass ceramic<br>Filler size: 0.01–0.1 μm | Voco, Cuxhaven, Germany |
| Estelite sigma quick | Supra-nano-filled | Matrix: Bis GMA, UDMA, TEGDMA<br>Filler: Silicon dioxide, Zirconium dioxide, Titanium dioxide<br>Filler size: 0.1–0.3 μm | Tokuyama Dental, Tokyo, Japan |

Note:
Bis-GMA, 2,2-bis[4-(2-hydroxy-3-methacryloxypropoxy)phenyl]propane; Bis-EMA, bis-phenol A ethoxylated dimethacrylate; UDMA, 1,6-bis(methacrylyloxy-2-ethoxycabonylamino)-2,4,4-trimethylhexane/urethane dimethacrylate; TEGDMA, triethyleneglycol dimethacrylate.

*Roselino et al., 2019*; *Vaz et al., 2019*; *Schlafer et al., 2021*; *Shang, Kaisarly & Kunzelmann, 2022*; *Kim et al., 2006*; *Jin et al., 2013*; *Yilmaz et al., 2021*; *Dos Santos et al., 2019*; *Celik & Iscan Yapar, 2021*). However, as far as we know, there has been no study comparing the effectiveness of n-HAP-containing toothpastes in removing the discoloration that may occur on the surfaces of composite restorative materials and their effects on the change in surface roughness with other whitening toothpastes. The null hypothesis tested is that whitening toothpaste including n-HAP will cause similar discoloration changes on the surfaces of different types of resin composites after different brushing times and will cause similar roughness changes on the surfaces of different types of resin composites at the end of the brushing procedure compared to other whitening toothpastes.

# MATERIALS AND METHODS

This study is an *in vitro* study using three different types of whitening toothpastes and three different types of composites.

## Preparation of samples and power analysis

*A priori* power analysis using WSSPAS web-based software revealed that the minimum sample size necessary to detect a significant difference was 25 in each group (75 in total), considering type I error (alfa) of 0.05, power (1-beta) of 0.9 and effect size of 1.05 (*Arslan et al., 2018*). The current study enrolled 90 composite materials (30 in each main material group) to increase the power of the current research. Disc-shaped specimens with a thickness of 2 mm and a width of 10 mm were prepared using microhybrid composite (3M Z250 Filtek; 3M ESPE, St. Paul, MN, USA), nanohybrid composite (Polofil, Voco, Cuxhaven, Germany), and supra-nano-filled composite (Estelite Sigma Quick, Tokuyama Dental, Tokyo, Japan) ($n = 30$). All preparation methods were carried out according to the manufacturer's instructions. The content of the composite materials is shown in Table 1.

Composite resins were placed in a specially prepared Teflon mold ($2 \times 10$ mm), and after the excess was removed by pressing lightly with transparent tape (Mylar strips), the samples were polymerized for 20 s with an LED light device (Guilin Woodpecker Medical Instrument Co., Guilin, China) emitting light at a wavelength of 470 nm (light intensity:

**Table 2 Whitening toothpastes used in the study.**

| Toothpaste | Compound | Manufacturer |
|---|---|---|
| Dentiste plus white | Sorbitol, Silicon Dioxide, Deionized Water, Xylitol, Glicerin, Green Tea Extract, Sodyum Lauryl Sulfate, PVP, Hydroxiapatite, Sodium Tripolyphosphate, Cellulose Gum, Titanium Dioxide, Sodium Benzoate, Sodium Saccharin, Menthol, Ascorbic Acid, Mica, Mentha Piperita Oil, Eucalyptus Globulus Leaf Oil, Eugenia Caryophyllus Flowe Oil, Salvia Officinalis Extract, Anthemis Nobilis Extract, Foeniculum Vulgare Extract, Glycyrrnhiza Grabra Extract, Cinnamomum Cassia Bark Extract, Cl 42090 | Dentiste, Thailand |
| Colgate optic white | Glycerin, Propylene Glycol, Calcium Pyrophosphate, PEG/PPG-116/66 Copolymer, PVP, PEG-12, Pentasodium Triphosphate, Sodium Lauryl Sulfate, Silica, Aroma, Sodium Monofluorophosphate, Disodium Pyrophosphate, Sodium Saccharin, Hydrogen Peroxide, Limonene, Cl74160 | Colgate-Palmolive, USA |
| Signal white system | Calcium Carbonate, Aqua, Sorbitol, Hydrated Silica, Sodium Lauryl Sulfate, Sodium Monofluorophosphate, Aroma, Cellulose Gum, Trisodium Phosphate, Perlite, Sodium Saccharin, Benzyl Alcohol, Glycerin, Sodium Laureth Sulfate, Cl74160 | Signal Unilever, UK |

**Note:**
PVP, polyvinylpyrrolidone; PEG, polyethylene glycol; PPG, polypropylene glycol.

1,200 mW/cm$^2$). Then, extra-coarse, coarse/medium, fine, and extra-fine polishing discs (OptiDisc, Kerr, FL, USA) were used consecutively for 15 s to polish the surfaces of the samples. New discs were used for each sample. All samples were stored in distilled water at 37 °C for 24 h.

The initial surface roughness (Ra$_0$) of each sample was measured using a mechanical profilometer (Mitutoyo SJ-210; Mitutoyo, Kawasaki, Japan), and basic color values were measured using a spectrophotometer (VITA Easyshade V, VITA Zahnfabrik, Bad Säckingen, Germany). Next, the samples were kept in a specially prepared coffee solution (Nescafe Gold, Nestle, Vevey, Switzerland) at room temperature for 5 days. The coffee solution was obtained by mixing 3 g of coffee powder and 150 ml of boiling water according to the manufacturer's instructions, and the solution was renewed daily. In previous studies, it was stated that the average daily coffee consumption in the population was 3.2 cups, and the average coffee consumption time was 15 min (*Yu et al., 2018*). The exposure time of 5 days, therefore, corresponded to approximately 5 months.

## Brushing procedure

Colored composite samples in three main material groups were divided into three subgroups ($n$ = 10): three different whitening toothpastes containing nanohydroxyapatite (Dentiste Plus White, Dentiste, Thailand), perlite (Signal White System, Signal Unilever, London, UK), and hydrogen peroxide (Colgate Optic White, Colgate-Palmolive, New York, NY, USA) were used to brush samples with an electric toothbrush (Oral B Clean DB04; Procter & Gamble, Cincinnati, OH, USA). The compositions of the toothpastes are listed in Table 2. Assuming that there are 28 teeth in the mouth, the maximum contact time of a tooth was determined as 10 s, based on the recommended 2 min of brushing twice a day (*Addy et al., 2002*). Thus, the groups were subjected to brushing simulation for 70, 150, and 300 s (7, 15, and 30 days, respectively). Each brushing session was performed by the same operator to ensure standardization.

## Surface roughness and color measurement

The final surface roughness ($Ra_1$) measurements were repeated at three different points of each sample with a mechanical profilometer (Mitutoyo SJ-210; Mitutoyo, Kawasaki, Japan), and average values were recorded. Color measurements of all samples were performed by the same operator using a spectrophotometer (VITA Easyshade V, VITA Zahnfabrik, Bad Säckingen, Germany) on a standard white background with the help of D65 illumination and CIEDE2000 color coordinates. The color coordinates of the samples were measured at the beginning of the brushing simulation (T0), after the coloring process (T1), at the end of the 7-day (T2), 15-day (T3), and 30-day (T4) brushing simulation applications. The samples were cleaned under running water after each brushing cycle, and color measurements were conducted immediately after washing. The CIEDE2000 formula was used to calculate the color changes (*Paravina et al., 2015*). The perceptibility threshold was determined as $\Delta E_{00} > 0.8$ units and the clinical acceptability threshold as $\Delta E_{00} \leq 1.8$ units (*Paravina et al., 2015*; *Tinastepe et al., 2021*).

$$\Delta E_{00} = \left(\frac{\Delta L'}{k_L S_L}\right)^2 + \left(\frac{\Delta C'}{k_C S_C}\right)^2 + \left(\frac{\Delta H'}{k_H S_H}\right)^2 + R_T \left(\frac{\Delta C'}{k_C S_C}\right)\left(\frac{\Delta H'}{k_H S_H}\right)$$

In the formula: $\Delta L'$, $\Delta C'$ and $\Delta H'$ denote the variations in lightness, chroma, and hue differences between the two samples. The weighting functions for the lightness, chroma, and hue parameters are $S_L$, $S_C$ and $S_H$. According to previous research, $K_L$, $K_C$ and $K_H$ are the regulated variables with regard to the many observed metrics (*Lee, 2005*; *Luo, Cui & Rigg, 2001*).

## Biostatistical analysis

The conformance of the quantitative data to the normal distribution was assessed with the Shapiro-Wilk test. Since the quantitative data did not show normal distribution ($p > 0.05$), the data were summarized with median and interquartile range (IQR). Kruskal-Wallis H test was employed for intergroup comparison of the data, and the Conover test was performed for *post-hoc* analysis. Wilcoxon signed-rank test was utilized to compare two repeated measurements. A two-way permutational multivariate analysis of variance (PERMANOVA) test was employed for color measurements related to the first factor (the groups: microhybrid composite-Dentiste Plus White, microhybrid composite-Colgate Optic White, microhybrid composite-Signal White System, nanohybrid composite-Dentiste Plus White, nanohybrid composite-Colgate Optic White, nanohybrid composite-Signal White System, supra-nano-filled composite-Dentiste Plus White, supra-nano-filled composite-Colgate Optic White, and supra-nano-filled composite-Signal White System) and the second factor (the measurements: $T_0$–$T_1$ ($\Delta E_{00}$), $T_1$–$T_2$ ($\Delta E1$), $T_1$–$T_3$ ($\Delta E2$) and $T_1$–$T_4$ ($\Delta E3$)). The PERMANOVA test was performed on Euclidean similarity matrices, and the residuals were achieved from a reduced model having 9,999 permutations. All $p < 0.05$ values were regarded as significant in all analyses. Multiple linear regression analysis with stepwise variable selection was performed to examine independent predictors (composites and kinds of toothpaste) of color, $Ra_0$, and $Ra_1$ results. The significance of multiple linear regression models was evaluated by analysis of variance

**Table 3 Color changes in $\Delta E_{00}$ ($T_0$–$T_1$) values in the main material groups.**

| Variable* | Material type | | | p-value# |
|---|---|---|---|---|
| | Microhybrid composite | Nanohybrid composite | Supra-nano-filled composite | |
| $\Delta E_{00}$ ($T_0$_$T_1$) | 5.248 (1.499) | 4.395 (2.277) | 4.158 (2.245) | 0.123 |

**Notes:**
* The data are summarized as 'median (interquartile range)'.
# Kruskal Wallis H test.

(ANOVA) test ($p < 0.05$), and the related statistics, including a 95% confidence interval of lower and upper bounds for the coefficients, R-squared, *etc.*, were also reported. Cohen d effect size was calculated to examine the clinical effects of significant differences for Ra measurements. In addition, box whisker plots were drawn for Ra and color results according to composite and toothpaste types. American Psychological Association (APA) 6.0 style was utilized to report the results of the statistical analyses (*Yagin et al., 2021*). All data analyses were carried out using PAST 4.10 and IBM SPSS Statistics 28.0 for Windows (New York, NY, USA).

# RESULTS

## Color change results

The color changes of the composite material groups after coloring with coffee for 5 days ($\Delta E_{00}$) are shown in Table 3. Among the main material groups, the group with the most coloration after exposure to coffee was the microhybrid composite group ($\Delta E_{00} = 5.24$). Color change in all materials exposed to coffee was clinically unacceptable ($\Delta E_{00} > 1.8$), and this change was not statistically significant ($p = 0.123$).

Table 4 shows the PERMANOVA test results of color changes between time periods of the subgroups. The distribution of the variables is given with a box-plot graph (Fig. 1). According to the findings; when the $\Delta E1$ values, which express the color change at the end of the 7th day, were examined, the most color change was seen with the Dentiste Plus White applied to the supra-nano-filled composite group. The least color change was observed after Signal White System applied to the microhybrid composite group. When the $\Delta E2$ values, which express the color change at the end of the 15th day, were examined, the most color change was seen after the Colgate Optic White toothpaste was applied to the supra-nano-filled composite group. The least color change was observed after Dentiste Plus White was applied to microhybrid composite group. When the $\Delta E3$ values, expressing the color change at the end of the 30th day were examined, the group with the most color change was the nanohybrid composite group in which Signal White System was applied. The least color change was seen in the nanohybrid composite group in which Dentiste Plus White was applied.

There were statistically significant differences among $\Delta E_{00}$, $\Delta E1$, $\Delta E2$ and $\Delta E3$ values in terms of color change values (F = 70.19, $p_1 < 0.001$). Color change measurements at $\Delta E_{00}$ were significantly higher than the measurements at $\Delta E1$, $\Delta E2$ $\Delta E3$ periods ($p < 0.001$). In the nanohybrid composite-Signal White System and supra-nano-filled

**Table 4 PERMANOVA test results for color change measurements.**

| Groups | Median (IQR) | Between measurements | Between groups | Interaction |
|---|---|---|---|---|
| | | F value | F value | |
| | | $p_1$ value | $p_2$ value | |
| MHC-DPW $\Delta E_{00}$ | 5.211 (1.4) | F = 70.19; | F = 2.46; | F = 1.17; |
| MHC-DPW $\Delta E1$ | 1.693 (1.067) | $p_1 < 0.001$ | $p_2 = 0.01$ | $p = 0.25$ |
| MHC-DPW $\Delta E2$ | 1.814 (0.423) | *Post-hoc* analyses | *Post-hoc* analyses | |
| MHC-DPW $\Delta E3$ | 2.751 (1.236) | (only significant pairwise comparisons) | (only significant pairwise comparisons) | |
| MHC-COW $\Delta E_{00}$ | 4.73 (1.118) | ($\Delta E_{00}$ and $\Delta E1$)* | (MHC-DPW and NHC-SWS)* | |
| MHC-COW $\Delta E1$ | 2.374 (1.071) | ($\Delta E_{00}$ and $\Delta E2$)* | (MHC-DPW and SNFC-DPW)* | |
| MHC-COW $\Delta E2$ | 2.53 (1.279) | ($\Delta E_{00}$ and $\Delta E3$)* | (NHC-DPW and NHC-SWS)* | |
| MHC-COW $\Delta E3$ | 2.975 (1.024) | ($\Delta E1$ and $\Delta E2$)* | | |
| MHC-SWS $\Delta E_{00}$ | 5.705 (1.698) | ($\Delta E1$ and $\Delta E3$)* | | |
| MHC-SWS $\Delta E1$ | 1.616 (0.716) | ($\Delta E2$ and $\Delta E3$)* | | |
| MHC-SWS $\Delta E2$ | 2.597 (0.811) | | | |
| MHC-SWS $\Delta E3$ | 3.35 (1.357) | | | |
| NHC-DPW $\Delta E_{00}$ | 4.226 (1.402) | | | |
| NHC-DPW $\Delta E1$ | 1.984 (1.682) | | | |
| NHC-DPW $\Delta E2$ | 2.875 (1.118) | | | |
| NHC-DPW $\Delta E3$ | 2.656 (0.839) | | | |
| NHC-COW $\Delta E_{00}$ | 4.405 (2.209) | | | |
| NHC-COW $\Delta E1$ | 2.262 (0.817) | | | |
| NHC-COW $\Delta E2$ | 2.955 (0.684) | | | |
| NHC-COW $\Delta E3$ | 3.17 (0.752) | | | |
| NHC-SWS $\Delta E_{00}$ | 5.409 (2.256) | | | |
| NHC-SWS $\Delta E1$ | 2.531 (1.045) | | | |
| NHC-SWS $\Delta E2$ | 2.473 (0.981) | | | |
| NHC-SWS $\Delta E3$ | 4.802 (2.044) | | | |
| SNFC-DPW $\Delta E_{00}$ | 4.246 (2.9) | | | |
| SNFC-DPW $\Delta E1$ | 2.986 (1.821) | | | |
| SNFC-DPW $\Delta E2$ | 2.197 (3.247) | | | |
| SNFC-DPW $\Delta E3$ | 3.64 (2.458) | | | |
| SNFC-COW $\Delta E_{00}$ | 3.525 (1.138) | | | |
| SNFC-COW $\Delta E1$ | 2.755 (1.84) | | | |
| SNFC-COW $\Delta E2$ | 3.3 (1.567) | | | |
| SNFC-COW $\Delta E3$ | 3.605 (1.681) | | | |
| SNFC-SWS $\Delta E_{00}$ | 4.596 (2.071) | | | |
| SNFC-SWS $\Delta E1$ | 1.892 (0.841) | | | |
| SNFC-SWS $\Delta E2$ | 2.476 (0.929) | | | |
| SNFC-SWS $\Delta E3$ | 3.595 (1.357) | | | |

**Notes:**

The data are summarized as 'median (interquartile range)'; IQR, interquartile range; $p_1$ value: significance test result between measurements, $p_2$ value; Intergroup PERMANOVA significance test result.

* Statistically significant ($p < 0.05$).

Color change values in the MHC-COW, microhybrid composite-colgate optic white; MHC-DPW, microhybrid composite-dentiste plus white; MHC-SWS, microhybrid composite-signal white system; NHC-COW, nanohybrid composite-colgate optic white; NHC-DPW, nanohybrid composite-dentiste plus white; NHC-SWS, nanohybrid composite-signal white system; SNFC-COW, supra-nano-filled composite-colgate optic white; SNFC-DPW, supra-nano-filled composite-dentiste plus white and SNFC-SWS, supra-nano-filled composite-signal white system composite-toothpaste subgroups at $\Delta E_{00}$: $T_0–T_1$; $\Delta E1$: $T_1–T_2$; $\Delta E2$: $T_1–T_3$; $\Delta E3$: $T_1–T_4$ times were detected and statistically compared.

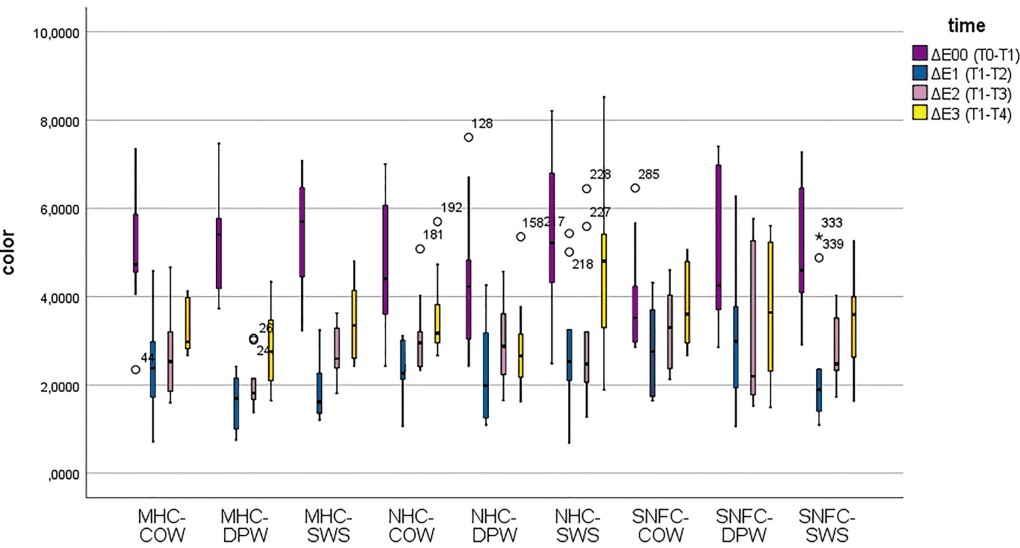

**Figure 1** Box-plot graphs for color change data of the MHC-COW, microhybrid composite-colgate optic white; MHC-DPW, microhybrid composite-dentiste plus white; MHC-SWS, microhybrid composite-signal white system; NHC-COW, nanohybrid composite-colgate optic white; NHC-DPW, nanohybrid composite-dentiste plus white; NHC-SWS, nanohybrid composite-signal white system; SNFC-COW, supra-nano-filled composite-colgate optic white; SNFC-DPW, supra-nano-filled composite-dentiste plus white and SNFC-SWS, supra-nano-filled composite-signal white system composite-toothpaste subgroups.

composite-Dentiste Plus White groups, color change measurements at $\Delta E1$ were significantly higher than color change measurements at $\Delta E2$, while color change measurements at $\Delta E1$ were significantly lower in all other groups compared to color change measurements at $\Delta E2$ ($p < 0.001$). Color change measurements at $\Delta E1$ were significantly lower than color change measurements at $\Delta E3$ ($p < 0.001$). While color change measurements at $\Delta E3$ were significantly lower in the nanohybrid composite-Dentiste Plus White group compared to color change measurements at $\Delta E2$, color change measurements at $\Delta E3$ were significantly higher in all other groups compared to color change measurements at $\Delta E2$ ($p < 0.001$). In addition, a statistically significant difference was found among the groups in terms of color change measurements (F = 2.46, $p_2 = 0.01$). Color change measurements were significantly different in the nanohybrid composite-Signal White System group compared to the microhybrid composite-Dentiste Plus White and nanohybrid composite-Dentiste Plus White groups ($p < 0.001$). Except for the color change measurements at $\Delta E_{00}$, color change measurements were significantly higher in the supra-nano-filled composite-Dentiste Plus White group compared to the microhybrid composite-Dentiste Plus White group ($p < 0.001$). According to the results obtained from the study, the time*group interaction effect for color change measurements was not found statistically significant (F = 1.17; $p = 0.25$).

In Table 5, the results of the multiple linear regression model created to determine the independent predictors of color change results are given. Since the stepwise variable selection method was used while creating the model, all variables were excluded from the model at the time of $\Delta E_{00}$. It was observed that microhybrid composite was a reducing

**Table 5 Multiple linear regression findings for color change results in each time period.**

| Time | Variable | $\beta$ | SE | t-value | p-value | 95% CI for $\beta$ | | R-squared |
|---|---|---|---|---|---|---|---|---|
| | | | | | | Lower | Upper | |
| $\Delta E1$ (T1–T2) | Constant | 2.646 | 0.152 | 17.374 | <0.001 | 2.343 | 2.948 | 0.067 |
| | Composite = Microhybrid | −0.665 | 0.264 | −2.523 | 0.013 | −1.190 | −0.141 | |
| $\Delta E2$ (T1–T3) | Constant | 3.051 | 0.139 | 22.014 | <0.001 | 2.776 | 3.326 | 0.061 |
| | Composite = Microhybrid | −0.575 | 0.240 | −2.395 | 0.019 | −1.052 | −0.098 | |
| $\Delta E3$ (T1–T4) | Constant | 3.681 | 0.153 | 24.018 | <0.001 | 3.376 | 3.985 | 0.046 |
| | Toothpaste = Dentiste plus white | −0.547 | 0.265 | −2.060 | 0.042 | −1.074 | −0.019 | |

**Note:**
SE, standard error; CI, confidence interval.

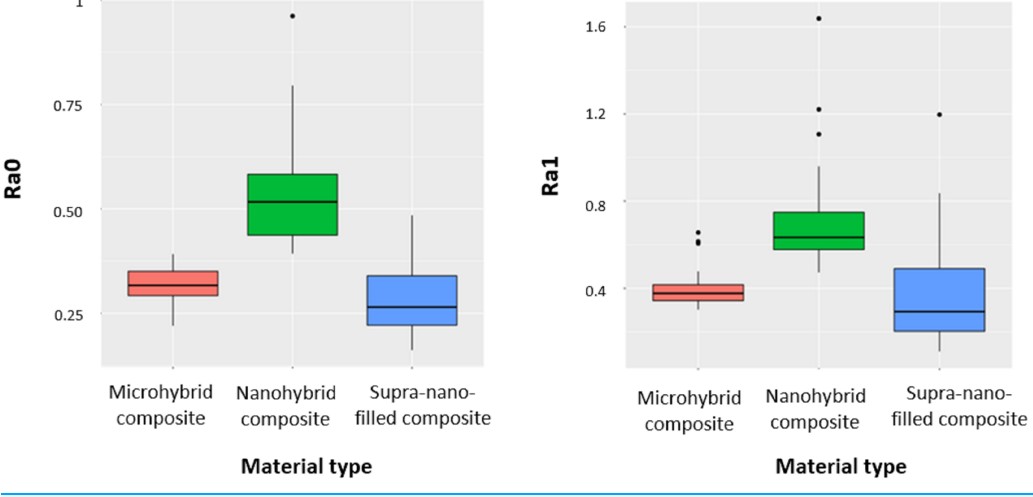

**Figure 2 Changes for $Ra_0$ and $Ra_1$ values in the main material groups.**

factor in the whitening effect of toothpastes at $\Delta E1$ (T1–T2) time ($\beta$ 95% CI −0.665 [−1.190 to −0.141]). Similarly, microhybrid composite was a reducing factor in the whitening effect of toothpastes at $\Delta E2$ (T1–T3) time ($\beta$ 95% CI −0.575 [−1.052 to −0.098]). At $\Delta E3$ (T1–T4) time, there was only Dentiste Plus White toothpaste in the model as a result of variable selection, and it was observed that Dentiste Plus White was a factor that had a reducing effect on color change results ($\beta$ 95% CI −0.547 [−1.074 to −0.019]).

## Surface roughness change results

The surface roughness findings of the main groups before ($Ra_0$) and after ($Ra_1$) brushing are shown in Fig. 2. According to the research findings, there was a statistically significant difference between the main groups in terms of $Ra_0$ and $Ra_1$ values ($p < 0.001$). $Ra_0$ and $Ra_1$ values were significantly higher in the nanohybrid composite group compared to the microhybrid composite and supra-nano-filled composite groups ($p < 0.001$). In addition, while the $Ra_0$ values were significantly higher in the microhybrid composite group compared to the supra-nano-filled composite group, no statistically significant difference

**Table 6 Changes in $Ra_0$ and $Ra_1$ values in composite-toothpaste subgroups.**

| Variable** | Groups* | | | | | | | | | $p$-value# | ES |
|---|---|---|---|---|---|---|---|---|---|---|---|
| | MHC-COW | MHC-DPW | MHC-SWS | NHC-COW | NHC-DPW | NHC-SWS | SNFC-COW | SNFC-DPW | SNFC-SWS | | |
| $Ra_0$ | $0.298^{ab}$ (0.036) | $0.338^c$ (0.024) | $0.302^{ab}$ (0.046) | $0.453^d$ (0.062) | $0.532^d$ (0.119) | $0.564^d$ (0.115) | $0.311^a$ (0.105) | $0.26^b$ (0.089) | $0.27^{ab}$ (0.152) | <0.001 | 2.82 |
| $Ra_1$ | $0.351^a$ (0.093) | $0.372^a$ (0.061) | $0.39^a$ (0.05) | $0.641^b$ (0.165) | $0.578^b$ (0.051) | $0.73^b$ (0.198) | $0.405^a$ (0.26) | $0.32^a$ (0.31) | $0.204^c$ (0.063) | <0.001 | 1.96 |
| $p$-value$ ($Ra_0$ vs. $Ra_1$) | 0.005 | 0.04 | 0.007 | 0.009 | 0.169 | 0.028 | 0.506 | 0.139 | 0.443 | | |

Notes:
* There is a statistically significant difference in-group categories that do not contain the same letter in each row (Conover test; $p < 0.05$).
** Variables are summarized as 'median (interquartile range)';
# Kruskal Wallis H test.
$ Wilcoxon signed-rank test. ES, effect size.
Initial ($Ra_0$) and final ($Ra_1$) surface roughness values in the MHC-COW, microhybrid composite-colgate optic white; MHC-DPW, microhybrid composite-dentiste plus white; MHC-SWS, microhybrid composite-signal white system; NHC-COW, nanohybrid composite-colgate optic white; NHC-DPW, nanohybrid composite-dentiste plus white; NHC-SWS, nanohybrid composite-signal white system; SNFC-COW, supra-nano-filled composite-colgate optic white; SNFC-DPW, supra-nano-filled composite-dentiste plus white and SNFC-SWS, supra-nano-filled composite-signal white system composite-toothpaste subgroups were detected and statistically compared.

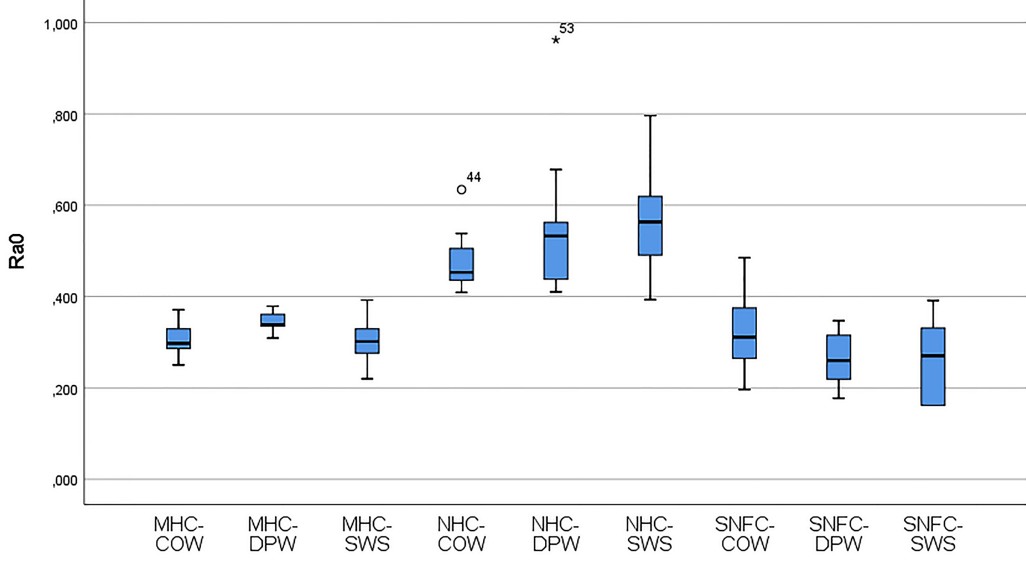

**Figure 3 Box-plot graphs for $Ra_0$ data of the MHC-COW, microhybrid composite-colgate optic white; MHC-DPW, microhybrid composite-dentiste plus white; MHC-SWS, microhybrid composite-signal white system; NHC-COW, nanohybrid composite-colgate optic white; NHC-DPW, nanohybrid composite-dentiste plus white; NHC-SWS, nanohybrid composite-signal white system; SNFC-COW, supra-nano-filled composite-colgate optic white; SNFC-DPW, supra-nano-filled composite-dentiste plus white; SNFC-SWS, supra-nano-filled composite-signal white system composite-toothpaste subgroups.**

was determined between the microhybrid composite and supra-nano-filled composite groups for the $Ra_1$ value.

Table 6 shows the variation of the $Ra_0$ and $Ra_1$ values of the materials according to the toothpaste subgroups. The distribution of the variables of the $Ra_0$ and $Ra_1$ values of the composite-toothpaste subgroups is given with a box-plot graph (Figs. 3 and 4). According

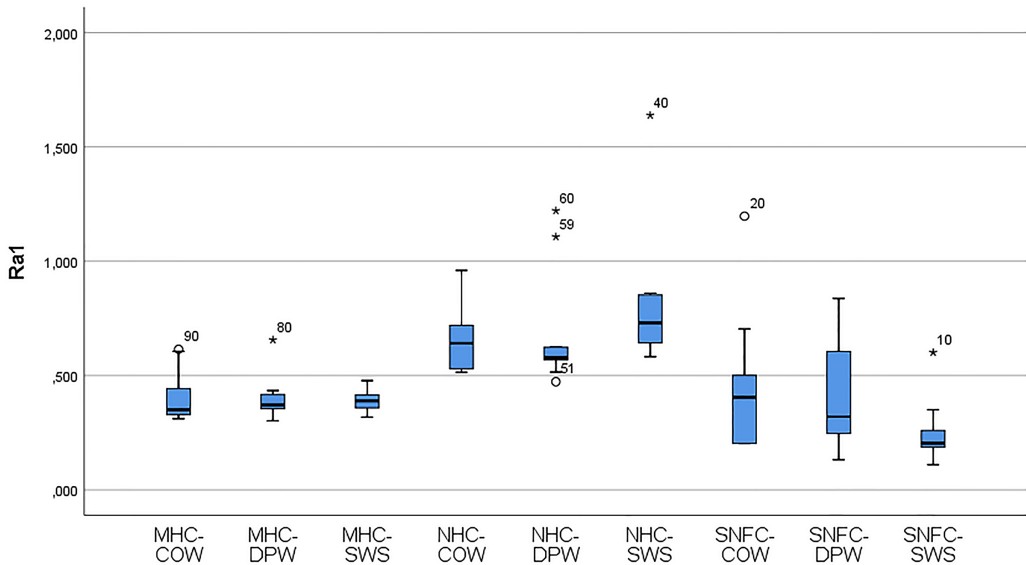

**Figure 4 Box-plot graphs for $Ra_1$ data of the MHC-COW, microhybrid composite-colgate optic white; MHC-DPW, microhybrid composite-dentiste plus white; MHC-SWS, microhybrid composite-signal white system; NHC-COW, nanohybrid composite-colgate optic white; NHC-DPW, nanohybrid composite-dentiste plus white; NHC-SWS, nanohybrid composite-signal white system; SNFC-COW, supra-nano-filled composite-colgate optic white; SNFC-DPW, supra-nano-filled composite-dentiste plus white; SNFC-SWS, supra-nano-filled composite-signal white system composite-toothpaste subgroups.**

**Table 7 Multiple linear regression findings for $Ra_0$ and $Ra_1$ results.**

| Model | Variable | $\beta$ | SE | t-value | p-value | 95% CI for $\beta$ | | R-squared |
|---|---|---|---|---|---|---|---|---|
| | | | | | | Lower | Upper | |
| $Ra_0$ | Constant | 0.301 | 0.012 | 25.397 | <0.001 | 0.278 | 0.325 | 0.590 |
| | Composite = Nanohybrid | 0.231 | 0.021 | 11.251 | <0.001 | 0.190 | 0.272 | |
| $Ra_1$ | Constant | 0.385 | 0.027 | 14.539 | <0.001 | 0.333 | 0.438 | 0.374 |
| | Composite = Nanohybrid | 0.333 | 0.046 | 7.253 | <0.001 | 0.242 | 0.424 | |

**Note:**
SE, standard error; CI, confidence interval.

to the findings of the study, there was a statistically significant difference between the composite-toothpaste subgroups in terms of $Ra_0$ and $Ra_1$ values ($p < 0.001$). $Ra_0$ and $Ra_1$ values were significantly higher in the nanohybrid composite-Signal White System group compared to the microhybrid composite-Colgate Optic White, microhybrid composite-Dentiste Plus White, microhybrid composite-Signal White System, supra-nano-filled composite-Colgate Optic White, supra-nano-filled composite-Dentiste Plus White, and supra-nano-filled composite-Signal White System groups (*Statistical significance expressed in APA style with letters in* Table 6). However, there was no statistically significant difference between nanohybrid composite-Colgate Optic White, nanohybrid composite- Dentiste Plus White, and nanohybrid composite-Signal White System groups in terms of $Ra_0$ and $Ra_1$ values (*Statistical significance expressed in APA style*

*with letters in* Table 6). In addition, the results of the repeated measurements for $Ra_0$ and $Ra_1$ values in the material-toothpaste subgroups indicated that $Ra_1$ values were significantly higher in microhybrid composite-Colgate Optic White ($p = 0.005$), microhybrid composite-Dentiste Plus White ($p = 0.04$), microhybrid composite-Signal White System ($p = 0.007$), nanohybrid composite-Colgate Optic White ($p = 0.009$) and nanohybrid composite-Signal White System ($p = 0.028$) groups compared to $Ra_0$ values.

In Table 7, the results of the multiple linear regression model created to determine the independent predictors of $Ra_0$ and $Ra_1$ results are given. In the results of $Ra_0$ (β 95% CI 0.231 [0.190 to 0.272]) and $Ra_1$ (β 95% CI 0.333 [0.242 to 0.424]), nanohybrid composite was observed to be a factor with an enhancing effect on surface roughness.

## DISCUSSION

Whitening toothpastes, including those containing nanohydroxyapatite (n-HAP), are frequently preferred by patients due to their easy accessibility and low cost compared to clinical applications. The effects of these materials on the discoloration and surface roughness of restoration surfaces are not well known. In this study, the effects of toothpastes containing n-HAP (Dentiste Plus White, Dentiste, Thailand), perlite (Signal White System, Signal Unilever, London, UK), and hydrogen peroxide (Colgate Optic White) on the surface roughness and color stability of nanohybrid, microhybrid, and supra-nano-filled composites were evaluated and compared. When the ΔE1 and ΔE2 values of n-HAP-containing whitening toothpaste (Dentiste Plus White, Dentiste, Thailand) were compared in the supra-nano-filled composite group, it showed a similar effect only in the nanohybrid composite group that was applied perlite-containing whitening toothpaste (Signal White System, Signal Unilever, London, UK) (ΔE1 > ΔE2). In the nanohybrid composite-Dentiste Plus White group, ΔE3 values caused less color change compared to ΔE2 values, unlike the other groups. When the results were evaluated in terms of roughness, none of the whitening toothpastes caused a significant increase in the surface roughness of the supra-nano-filled composite group. Whitening toothpaste containing n-HAP did not cause a significant increase in the surface roughness of both the supra-nano-filled composite group and the nanohybrid composite group. Therefore, the null hypothesis of the present study was partially accepted.

The CIEDE2000 formula was used to evaluate the color changes in the study. This formula is a good indicator of human perceptible and acceptable color changes (*Paravina, Pérez & Ghinea, 2019*). Color changes were evaluated by making comparisons with 50:50% perceptibility and 50:50% acceptability thresholds. The perceptibility threshold for CIEDE2000 was taken as $\Delta E_{00} > 0.8$ units and the clinical acceptability threshold as $\Delta E_{00} \leq 1.8$ units (*Paravina et al., 2015*). The color of esthetic restorative materials can change over time due to external factors. Roughness and discoloration on the surface of restorative materials and tooth lead to negative esthetic results (*Rode et al., 2021*; *Yilmaz et al., 2021*). Studies have emphasized that red wine, coffee, and tea are important external factors (*Celik & Iscan Yapar, 2021*; *Omata et al., 2006*; *Temizci & Tunçdemir, 2021*). In studies (*Omata et al., 2006*; *Turker, Kocak & Aktepe, 2006*; *Farawati et al., 2019*; *Quek et al., 2018*), coffee, tea, cola, and red wine were used as coloring solutions, while only coffee was used in others

(*Celik & Iscan Yapar, 2021*; *Temizci & Tunçdemir, 2021*; *Turgut et al., 2018*). While *Turker, Kocak & Aktepe (2006)* found that red wine and tea were the beverages that produced the most coloration, *Farawati et al. (2019)* used wine, coffee, and tea as a coloring solution in their study, and coffee and wine produced significantly more coloration than tea, but did not show any significant difference from each other. According to some authors, coffee causes the most discolouration, followed by red wine and tea (*Paolone et al., 2022*). *Manno et al. (2018)* have identified coffee as the most popular beverage consumed by millions of people around the world every day. Therefore, the coffee solution was used as a colorant in the present study. The critical coloration period is the first 5 weeks, during which stain penetration can reach up to five microns (*Ozkanoglu & Akin, 2020*). In this study, the samples were immersed in coffee for 5 days (*Celik & Iscan Yapar, 2021*). In the $\Delta E_{00}$ data obtained after coloring with coffee, color changes occurred in all groups above the clinically acceptable threshold. When the data obtained after the application of whitening toothpastes were evaluated, no color change occurred above the clinically acceptable threshold in terms of $\Delta E1$ values only in the microhybrid composite-Dentiste Plus White and microhybrid composite-Signal White System groups. The adsorption and absorption mechanisms occurring in the organic phase of the composite explain the coloration that occurs with coffee (*Celik & Iscan Yapar, 2021*). As a result of our study, the group that showed the most coloration after exposure to coffee was the microhybrid structured composite group, while the group that showed the least coloration was the supra-nano-structured composite group. In their study evaluating the color stability of microhybrid and nanohybrid composites, *Ugurlu, Temel & Hepdeniz (2019)* stated that the most colored samples in all subgroups belonged to the microhybrid composite group. In another study, *Aydın et al. (2021)* investigated the surface coloration changes of composite samples containing different types of fillers with polished surfaces. As a result, as in our study, the most colored group was the composite group containing microhybrid particles.

The coloration differences between composites are closely related to the resin matrix and filler type (*Erdemir, Yildiz & Eren, 2012*). In their study examining the effects of various factors on the color of composites, *Menon, Ganapathy & Mallikarjuna (2019)* stated that the large filler particle size of the composite and the fact that the filler content is composed of different materials are important factors that increase the coloration. The fact that the microhybrid composite contains larger particles in the inorganic part compared to the other composites used in the study may be one of the factors explaining its higher coloration with coffee and lesser color change after the application of whitening toothpastes.

There are many whitening toothpastes on the market to remove or prevent discoloration on tooth surfaces. It is thought that substances such as calcium carbonate, perlite, calcium pyrophosphate, silicon dioxide, and hydrogen peroxide in the paste provide a whiter appearance by removing the chromophore and biofilm that cause coloration (*Koc Vural et al., 2021*). An innovative tooth-whitening component is hydroxyapatite (HAP) (*Limeback, Meyer & Enax, 2023*). *In vivo* and *in vitro* studies using HAP have generally investigated its remineralizing and desensitizing effects. Studies examining the teeth-whitening effect of HAP are limited. *Shang, Kaisarly & Kunzelmann*

(2022) stated that, as a result of their *in vitro* study, the addition of n-HAP particles to toothpaste provided effective whitening. In their observational clinical study, *Steinert et al. (2020)* examined the effects of an oral gel containing microcrystalline hydroxyapatite on teeth whitening over the course of 4 weeks of twice-daily dental care. They claimed that the biomimetic alternative to existing whitening agents for routine dental care, microcrystalline hydroxyapatite, is a viable whitening ingredient for oral care formulations. *Steinert et al. (2020)* examined the impact of a toothpaste containing biomimetic zinc hydroxyapatite (HAP) on subjective metrics during a four-week at-home use in another observational trial. Patients in this study stated that using the HAP toothpaste resulted in smoother and whiter teeth. In a randomized, double-blinded controlled clinical trial (*Woo et al., 2014*), 85 individuals with tooth discolorations were randomly selected to use one of three toothpastes containing either hydroxyapatite, hydrogen peroxide, or no active ingredient (placebo toothpastes). The patients were examined at the start of treatment and 1, 2, and 3 months later. This study found that toothpaste containing hydrogen peroxide was significantly more effective at lightening teeth than hydroxyapatite and placebo toothpastes (*Woo et al., 2014*). In our study, Dentiste Plus White toothpaste containing silicon dioxide and n-HAP showed the highest difference in color change analyzed at the end of the first brushing simulation after coloring with coffee ($\Delta E1$). This change was found to be significantly higher in the supra-nano-filled composite-Dentiste Plus White group than in the microhybrid composite-Dentiste Plus White group at all brushing time intervals. In the supra-nano-filled composite-Dentiste Plus White group, the color change value obtained after 7 days of brushing ($\Delta E1$), which included the first brushing performed after coloring with coffee, was also higher than the value obtained after brushing completed after 15 days ($\Delta E2$). These findings show that n-HAP in toothpaste can provide more effective whitening in the early period. In terms of $\Delta E2$ values, the group that showed the most color change was Colgate Optic White toothpaste containing silicon dioxide and hydrogen peroxide. The group with the highest color change in terms of $\Delta E3$ values was the nanohybrid composite group in which Signal White System containing perlite was applied. In addition, the nanohybrid composite-Signal White System group showed a significant difference in color change measurements compared to the microhybrid composite-Dentiste Plus White and nanohybrid composite-Dentiste Plus White groups. Perlite, which is widely used in whitening toothpastes, has been highlighted for its good stain removal and polishing properties (*Joiner et al., 2002*). According to the results of our study, the highest whitening effect was obtained with toothpastes containing silicon dioxide (Dentiste Plus White and Colgate Optic White) on days 7 and 15 of measurement, while the highest color change was observed only on day 30 with perlite-containing toothpaste. In another study, *Joiner (2009)* stated that silicon dioxide-based toothpastes have whitening properties, and this effectiveness can be increased with other whitening agents that can be added to the toothpaste. The prominent color change of Dentiste Plus White and Colgate Optic White containing silicon dioxide as a common material at different times may be due to the different activities of n-HAP and hydrogen peroxide, respectively, which are also part of their contents.

In our study, the restorative material with the least color change in terms of ΔE1 and ΔE2 values was a microhybrid composite, while the material with the most color change was a supra-nano-filled composite. It is thought that the different results of toothpastes on different restorative materials are related to the results of their interactions with the composites, as well as the abrasives contained in the toothpastes. Concerning this, *Monaghan, Trowbridge & Lautenschlager (1992)* reported that the formulations of composite resins were different, resulting in different degrees of whitening. They also stated that the rate of conversion of the resin matrix to polymer allowed some resins to be more exposed to the bleaching material, while others were less affected.

When the surface roughness values of the composites before and after brushing were examined, the surface roughness of all the samples increased in general. Toothpastes caused significant roughness in the microhybrid composite. The supra-nano-filled composite was not significantly affected by the toothpastes in terms of roughness. The reason toothpastes cause different surface roughness increases on composite surfaces is related to the structural differences of composites (*Menon, Ganapathy & Mallikarjuna, 2019*). Resin composites with different formulations have different surface roughness resistances (*Hu, Marquis & Shortall, 2003*). The friction and shear forces that occur on the surface of the restorative material as a result of brushing cause material loss. The roughest surface before and after brushing belonged to the nanohybrid composite group. The fact that the composite with the nanohybrid particle structure is rougher than the microhybrid composite with larger particles may depend on the type, shape, and degree of hardening of the inorganic fillers (*Ruivo et al., 2019*). *Farghal & Elkafrawy (2020)* obtained a similar result in their study: they found the surface roughness values of the microhybrid composite to be lower than those of the nanohybrid composite. They stated that due to the production technique, the microhybrid composite did not undergo inorganic particle separation and elution during polishing, and that large particles could be broken off from the nanohybrid composite.

This study has some limitations. One of them was that the coloring was created only with coffee. Although coffee is the most consumed beverage in the world, the effect of other external coloring factors should also be evaluated. Another limitation is that oral conditions (food or drinking habits, temperature changes, saliva, smoking, *etc.*) cannot be directly imitated, because the results of the study were obtained by *in vitro* experiments. To subject all materials to the same technique and avoid any potential bias due to differences in saliva structure, food or drinking habits, and so on, we chose to conduct *in vitro* research in standardized laboratory settings. This method provided comprehensive control over all variables. Furthermore, this technique has a clear ethical benefit because it does not put humans in danger. Studies are needed to evaluate the discoloration and surface roughness effects of whitening toothpastes containing n-Hap on colored restorations *in vivo*.

## CONCLUSIONS

Within the limitations of the present *in vitro* study, it can be concluded that while whitening toothpastes containing nanohydroxyapatite and hydrogen peroxide provided a

high color change in the early period, toothpaste containing perlite provided a higher color change over a longer time. In addition, while none of the whitening toothpastes showed a significant increase in the roughness of the supra-nano-filled composite surface, the toothpaste containing nanohydroxyapatite did not cause a significant increase in the surface roughness of the nanohybrid composite either. The use of whitening toothpastes containing nanohydroxyapatite can provide a color change without creating significant roughness on the surface of nanohybrid and supra-nano-filled restorative materials in cases where extrinsic discoloration is expected to be removed within a short time.

### Funding
The authors received no funding for this work.

### Competing Interests
The authors declare that they have no competing interests.

### Author Contributions

- Burak Dayı conceived and designed the experiments, analyzed the data, prepared figures and/or tables, authored or reviewed drafts of the article, and approved the final draft.
- Fikri Öcal conceived and designed the experiments, performed the experiments, prepared figures and/or tables, authored or reviewed drafts of the article, and approved the final draft.

### Data Availability

The raw data of measurements are available in the Supplemental Files.

### Supplemental Information

Supplemental information for this article can be found online at http://dx.doi.org/10.7717/peerj.15692#supplemental-information.

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
