# Peer review of "Evaluation of the effects of whitening toothpaste containing nanohydroxyapatite on surface roughness and color change in restorative materials"

_PeerJ, doi:10.7717/peerj.15692_

## Round 0.1 · original submission · Major Revisions

If the comments raised by the reviewers can be addressed I will be willing to reconsider the paper. Please add an extensive point-to-point reply to each reviewer.

·

Basic reporting

Tucker et al (2006) found that red wine and tea caused the most significant colour changes of composite resin
Please explain why the authors used coffee in this study?

Note:
Turker SB, Kocak A, Aktepe E. Effect of five staining solutions on the colour stability of two acrylics and three composite resins based provisional restorations. Eur J Prosthodont Restor Dent. 2006;14:121–5

Experimental design

The authors wrote in Results section:
Color change in all materials exposed to coffee was clinically unacceptable and this change was not statistically significant (p = 0.123).

That statement mean: there is no different of color change between all composite resin used in this study.

Could you explain why that condition happened?
We know that filler particle size of the composite resin will affect the coloration of composite resin.

Validity of the findings

Please state your conclusion clearly/briefly based on your data/findings.

Additional comments

-

Reviewer 2 ·

Basic reporting

The study's use of unambiguous and professional English is commendable. However, while the literature references are sufficient, the field background and context are not adequately provided, which may hinder readers' understanding. The article structure is professional, with well-designed figures, tables, and raw data shared. Nevertheless, the study is not entirely self-contained, as some relevant results to the hypotheses are missing.

Experimental design

The study represents original primary research that falls within the Aims and Scope of the journal. However, the research question is not well-defined, and it is unclear how the investigation addresses a relevant and meaningful knowledge gap. Additionally, the study has certain flaws that require attention. On a positive note, the methods employed are described with sufficient detail and information to enable replication.

Validity of the findings

The study has provided all underlying data, but they are not robust, statistically sound, or well-controlled. Despite this limitation, the conclusions drawn from the data support the results obtained.

Additional comments

The present study aimed to evaluate the effects of a whitening toothpaste containing nanohydroxyapatite on the surfaces of restorative materials colored with coffee compared to those containing perlite and hydrogen peroxide. Although the manuscript is well-written, I cannot recommend it for publication due to some limitations, which are listed below:

1)Limited scope: The methodology is limited to only testing the effects of brushing with three different types of toothpaste on the surface roughness and color of composite resin specimens, which does not provide a comprehensive evaluation of other factors that could affect composite restorations in vivo.

2) Artificial conditions: The specimens were prepared in a laboratory setting and were not exposed to the full range of environmental factors and stresses that would be encountered in the oral cavity, such as temperature changes, saliva, and bacterial biofilm.

3) Small sample size: The sample size of 30 specimens for each composite material group is relatively small, which could limit the generalizability of the findings.

4) Lack of long-term evaluation: The study evaluated the effects of toothpaste on restoration surfaces only for a short-term period of 15 days, which means the long-term effects of toothpaste on restoration surfaces are not known.

5) Lack of external validity: The study is an in vitro study, which means that the results may not translate well to real-life situations. The conditions under which the samples were tested may not mimic those of the oral cavity.

6) Limited clinical relevance: The study only assessed color changes and surface roughness of composite resin specimens, without considering other clinical parameters such as wear, marginal integrity, or secondary caries formation, which are important for the long-term success of restorations.

7) Single colorant used: The study used only one colorant (coffee) for evaluating the color changes in the restoration surfaces. Other external factors that can cause discoloration, such as red wine and tea, were not considered.

8) Simplified brushing simulation: The brushing simulation used in this methodology only considers a maximum contact time of 10 seconds per tooth, which may not reflect the actual brushing habits of individuals, and therefore may not provide an accurate representation of the effects of toothbrushing on composite restorations in vivo.

9) Non-standardized brushing: The brushing simulation was not standardized, which may have affected the results of the study.

10) Lack of control group: The study does not include a control group that was not subjected to brushing simulation, making it difficult to determine the effects of brushing alone versus the effects of the toothpastes used.

11) Lack of evaluation of other external factors: The study has only evaluated the effects of coffee on the color stability of composites, while other external factors such as smoking, food, and beverages have not been considered.

12) Lack of clinical data: The study did not provide any clinical data on the efficacy of the tested toothpastes in removing tooth discoloration.

---

## Round 0.2 · accepted · Accept

Best wishes to the authors and thank you for addressing to the comments raised.

·

Basic reporting

Clear, after revision introduction more adequate.
The authors explained used coffee in this study

Experimental design

Clear, methods described with adequate detail and information

Validity of the findings

The authors explained why p=0,123

Additional comments

-